# Effect of Jump Direction on Joint Kinetics of Take-Off Legs in Double-Leg Rebound Jumps

**DOI:** 10.3390/sports7080183

**Published:** 2019-07-26

**Authors:** Yasushi Kariyama

**Affiliations:** Faculty of Sport Science, Yamanashi Gakuin University, Kofu 400-8575, Japan; y-kariyama@ygu.ac.jp; Tel.: +81-55-224-1319

**Keywords:** plyometric training, stretch-shortening cycle, joint power, inverse dynamics

## Abstract

Vertical (VDJ) and horizontal (HDJ) double-leg rebound jumps are used as plyometric exercises in direction-specific training regimens for various sports. We investigated the effects of jump direction on joint kinetics of the take-off legs in double-leg rebound jumps. Twelve Japanese male track and field athletes performed VDJ, 100% HDJ, 50% HDJ (50% of 100% HDJ distance), and 75% HDJ (75% of 100% HDJ distance). Kinematic and kinetic data in the sagittal plane were calculated using a force platform and high-speed video camera. Hip negative power during the eccentric phase decreased from VDJ to 50% HDJ (VDJ, −4.40 ± 4.25 W/kg; 50% HDJ, −0.83 ± 2.10; 75% HDJ, −0.33 ± 0.83; 100% HDJ, 0 ± 0), while hip positive power increased from VDJ to 100% HDJ (VDJ, 4.19 ± 2.73 W/kg; 50% HDJ, 9.37 ± 2.89; 75% HDJ, 11.15 ± 3.91; 100% HDJ, 18.51 ± 9.83). Knee negative power increased from VDJ to 75% HDJ (VDJ, −14.48 ± 7.67 W/kg; 50% HDJ, −18.98 ± 7.13; 75% HDJ, −21.57 ± 8.54; 100% HDJ, −23.34 ± 12.13), while knee positive power decreased from VDJ to 75% HDJ (VDJ, 23.18 ± 9.01 W/kg; 50% HDJ, 18.83 ± 5.49; 75% HDJ, 18.10 ± 5.77; 100% HDJ, 16.27 ± 6.22). Ankle joint kinetics remained unchanged. Differences in hip and knee joint kinetics between VDJ and HDJ were associated with direction control, becoming more pronounced as jump distance increased.

## 1. Introduction

Plyometric training employing jumping exercises is widely used to enhance the mechanical output of the lower limbs in running and jumping by improving muscle and tendon function [1,2]. Typical jumping exercises consist of bounce-type or rebound-type double-leg jumping in the vertical direction, such as a drop jump or repetitive rebound jump, which is termed a vertical double-leg rebound jump (VDJ). As multiple athletic activities are performed in the horizontal direction, the horizontal rebound jump (including the horizontal double-leg rebound jump (HDJ)) is often used for plyometric training (e.g., a drop jump in the horizontal direction and the hurdle jump) as part of direction-specific training regimens [2,3,4,5].

The outcomes of plyometric training are influenced by the characteristics of the exerted muscle power [6,7]. Therefore, identifying the differences in the joint kinetics of the take-off legs between HDJ and VDJ could aid in the development of more effective and specific training methods. Multiple studies have investigated VDJ kinetics [6,8,9,10]; VDJ is a beneficial plyometric exercise for improving the power output of the ankle plantar flexors. Although multiple studies have investigated the kinematics and ground reaction force of HDJ [11,12], very few studies have investigated joint kinetics [13]. To my knowledge, no studies have investigated the differences in the joint kinetics between HDJ and VDJ. Therefore, it remains unclear if the kinetics are altered based on the jump direction during the double-legs rebound jump.

For plyometric training using HDJ, the training intensity is modified by changing the jump distance [2,3,4,5]. One of the most important variables affecting training outcomes during HDJs is the jump distance, which is determined by the joint kinetics of take-off [14]. Therefore, different jump distances during HDJs would produce different joint kinetics in take-off legs; thus, different plyometric training outcomes are expected for HDJs at different jump distances. However, to my knowledge, no studies have investigated the effect of jump distance during the HDJ. This information, if available, could help elucidate potential differences in kinetics based on jump direction, and thus facilitate efforts to more effectively tailor plyometric exercises for sport based on type and the training needs of athletes.

The aim of this study was to investigate the effect of jump direction on the joint kinetics of the take-off legs during double-leg rebound jumps and to elucidate the advantages of the HDJ as a mode of plyometric training. Differences have been shown between standing horizontal and vertical jumps using a double-leg take-off with counter movement [15,16,17]. Compared to the vertical jump, the horizontal jump employs the hip joint as the main energy generator [17], and the lower knee joint kinetics controls the jump direction [15,16]. Moreover, during sprint running in the horizontal direction, peak hip extension torque [18,19] and power [18] increase as running speed increases. Furthermore, as faster sprinters show less knee extension during the support phase; knee extension is thus unnecessary to achieve a high sprinting speed [20]. The present study hypothesized that differences in joint kinetics between HDJ and VDJ were observed in the hip and knee joints based on the jump direction. Moreover, this would become more pronounced with an increase in jump distance during HDJ.

## 2. Materials and Methods

### 2.1. Participants

Twelve Japanese male track and field sprinters and jumpers (age, 22.0 ± 2.2 years; height, 1.75 ± 0.61 m; mass, 65.80 ± 4.02 kg) participated in this study. Informed written consent was obtained from each participant prior to study participation. All trained athletes were members of a university track and field team as sprinters and jumpers. They had performed regular sprint and jump training, including experimental tasks 5–6 days per week for more than 7 years. All participants competed from the regional to international level within the preceding year and were familiarized with the experimental trials. All procedures undertaken in this study were conducted in accordance with the Declaration of Helsinki and approved by the relevant institutional ethics committee (ethical code: 21-399).

### 2.2. Experimental Procedures

Data were collected at an outdoor track and field stadium. To ensure that every individual received adequate warm-up, the individuals were free to follow their own style and duration of warm-up (consisting of jogging, static and dynamic stretching, and various jumps) for approximately 30–60 min. After a warm-up period, the participants performed VDJ, 50% HDJ, 75% HDJ, and 100% HDJ (Figure 1). Each jump was performed at least twice. Jumps consisted of six repeated rebound jumps in the vertical or horizontal direction with a double-leg take-off from a standing posture. For VDJ, the participants were instructed to maintain the contact time as short as possible and to jump as high as possible [10,21]. The 100% HDJ commenced from a double-leg standing position, and the participants were instructed to maintain the contact time as short as possible and to jump as far as possible, by performing a series of six forward double-leg jumps. In the 50% HDJ and 75% HDJ, the markers were set at equal intervals of 50% and 75% of jump distance in 100% HDJ, respectively. In 50% HDJ and 75% HDJ, participants were instructed to maintain the contact time as short as possible and to jump as high as possible between the markers by performing a series of six forward double-leg jumps. Based on previous testing sessions, the starting mark for each horizontal jump (50% HDJ, 75% HDJ, and 100% HDJ) was used to allow the participants to strike a force plate at the fifth step without altering their technique immediately before reaching the plate. All participants wore personal training shoes (without spikes) during the experiment. Participants rested as often as they felt necessary between trials to reduce the effects of fatigue.

Trials with the highest reactive strength index for all jumps were selected for further analysis. A successful trial was defined by the participant landing within the boundaries of any force platform on the fifth step of the 50% HDJ, 75% HDJ, and 100% HDJ, and any landing for the VDJ.

### 2.3. Data Analysis

The jumping motions of participants were recorded in the sagittal plane using a high-speed video camera (EX-F1, 300 fps; Casio, Tokyo, Japan) positioned at the center of the track lane to the right of the 40 m mark. Ground reaction force was obtained using one 3D force platform (9287B 0.9 m × 0.6 m; Kistler Instrumente AG, Winterthur, Switzerland; 1000 Hz) for VDJ, 50% HDJ, and 75% HDJ, and three 3D force platforms (9287B, 0.9 m × 0.6 m; 9281A, 0.6 m × 0.4 m; 9281C, 0.6 m × 0.4 m; 1000 Hz) for 100% HDJ connected to a single computer (DXP061; Dell, Tokyo, Japan).

A 14-segment model used by Ae [22] comprising the hands, forearms, upper arms, feet, lower legs, thighs, head, and trunk was used. Reference markers were affixed to each body segment. Twenty-three body points (hands, wrists, elbows, shoulders, toes, first metatarsal bones, heels, ankles, knees, greater trochanters, head, ears, and the suprasternal area) and four calibration markers were manually digitized using a Frame-DIAS system (DKH Co., Tokyo, Japan), starting at 10 frames prior to touchdown and ending at 10 frames after toe-off. A single skilled digitizer with nearly 10 years of experience performed the digitizing procedure. All subsequent data analyses were performed using MatlabTM (v. 8.3.0, The MathworksTM; Natick, MA, USA). The raw coordinates were converted to real coordinates using four reference markers placed on the ground. The coordinates were smoothened using a Butterworth digital filter with optimal cut-off frequencies of 7.5–10.5 Hz and determined using the residual method [23]. For the kinetic inputs to inverse dynamic analysis, force platform data (1000 Hz) were down-sampled to the sampling rate of the kinematic data (300 Hz).

Reactive strength index was calculated by dividing the jump height by the contact time [10,21,24,25]. Contact time and flight time were calculated using the vertical ground reaction force data (the threshold of the ground reaction force level was based on 3% of the body mass). Jump height was calculated using Bosco’s theory [26]. The take-off phase (from the point of touchdown to toe-off) was divided into two parts as follows: the eccentric (from the point of touchdown to the lowest point of the body’s center of gravity) and concentric (from the lowest point of the body’s center of gravity to toe-off) phases.

During the jumps, the location of the center of mass and inertia of each segment were estimated based on the body segment parameters for Japanese athletes [22]. Joint torques were calculated using an inverse dynamics approach (Figure A1 in Appendix A). Joint power was calculated as the dot product of joint torque and angular velocity. Extension and plantar flexion were denoted as positive at each of the three leg joints. To evaluate joint torque and power output characteristics for a single leg during the rebound jump, the ground reaction force was divided in half, and this data was used to calculate the joint kinetic parameters. In addition, the peak ankle, knee, and hip joint torque (eccentric and concentric phases) and the peak ankle, knee, and hip joint power (negative and positive values) were calculated using plantar flexion and/or extension joint torque. The time series data were normalized to the time of the take-off phase (0–100%).

### 2.4. Statistics

Descriptive statistics are presented as mean values ± standard deviation. Each variable was compared by a one-way analysis of variance with repeated measures. The magnitude of effect was calculated using partial η^2^. Values of 0.04, 0.25, and above 0.64 were considered small, medium, and large, respectively [27]. Statistical analysis was executed using SPSS (IBM SPSS Statistics Version 24, SPSS; Chicago, IL, USA). The two-tailed paired t-test (adjusted by Holm’s method) was used to determine differences between jump type for each dependent variable. The interclass correlation coefficient (one-way random, single measure) for the reactive strength index for the VDJ and 100% HDJ over two successful trials was calculated. The Shapiro-Wilk test was used to test for normal distribution of the reactive strength index in the VDJ and 100% HDJ. Statistical significance was set at α < 0.05.

## 3. Results

Interclass correlation coefficients for reactive strength index for the VDJ and that for 100% HDJ were 0.969 and 0.872, respectively (*p* < 0.001). Moreover, these values had normal distribution as per the Shapiro-Wilk test (VDJ, *p* = 0.323; 100% HDJ, *p* = 0.760).

The reactive strength index was 3.469 ± 0.574 m/s for the VDJ. As jump distance increased, horizontal velocity increased and vertical velocity decreased; however, contact time was unchanged (Table 1, Table A1 in Appendix B; contact time, F(3,33) = 0.119, *p* = 0.949; vertical velocity; F(3,33) = 1653.365, *p* < 0.001; horizontal velocity, F(3,33) = 23.188, *p* < 0.001).

For the hip joint, as jump distance increased, peak negative power significantly decreased, whereas peak positive power increased (Table 2, Table A1 in Appendix B; ecc torque, F(3,33) = 6.066, *p* = 0.002; con torque, F(3,33) = 1.132, *p* = 0.350; negative power, F(3,33) = 9.416, *p* < 0.001; positive power, F(3,33) = 15.687, *p* < 0.001). For the knee joint, peak negative power for VDJ was smaller than that for 75% HDJ and 100% HDJ, whereas peak positive power for VDJ was higher than that for 75% HDJ and 100% HDJ (ecc torque, F(3,33) = 0.686, *p* = 0.567; con torque, F(3,33) = 1.580, *p* = 0.213; negative power, F(3,33) = 4.868, *p* = 0.007; positive power, F(3,33) = 6.007, *p* = 0.002). For the ankle joint, peak negative power for 50% HDJ and 75% HDJ were smaller than that VDJ. However, other ankle joint kinetics did not vary and the magnitude of effect for almost all of the ankle joint kinetics was small, and smaller than that for other joints (ecc torque, F(3,33) = 0.556, *p* = 0.648; con torque, F(3,33) = 0.211, *p* = 0.888; negative power, F(3,33) = 1.062, *p* = 0.379; positive power, F(3,33) = 0.244, *p* = 0.865). Patterns for ankle joint parameters during VDJ were similar to those during the HDJs (Figure 2). However, the hip joint showed almost no flexion during the HDJs, whereas hip joint flexed during early take-off phase in the VDJ. Hip extension torque during the early take-off phase (approximately 0–20%) in the HDJs was higher and occurred earlier than that in VDJ, and was more pronounced as jump distance increased. The timing exerted for knee joint flexion torque (at approximately 80% of take-off phase) was earlier, and extension torque during knee extension (between approximately 40–80% of the take-off phase) was smaller. Both knee flexion and extension torque were more pronounced with increasing jump distance.

## 4. Discussion

In this study, I found that as jump distance increased, hip negative power decreased, and hip positive power increased. However, knee negative power increased during 75% HDJ, whereas knee positive power decreased during 75% HDJ. Therefore, these results support my hypothesis that differences in joint kinetics between HDJ and VDJ would be observed in the hip and knee joints and would change depending on the jump direction. Furthermore, these became pronounced as the jump distance increased.

Regarding the hip joint, as the jump distance increased, negative power decreased, and positive power increased (Table 2). Moreover, negative power and flexion velocity approached zero (Table 2, Figure 2). These results indicate that the primary function of the hip extensor may be as a mechanical energy generator for the HDJs performed in this study, and that this function in the hip is especially augmented with increases in the jump distance. For horizontal jumps using a double-leg take-off with counter movement, the hip joint is the main energy generator in the horizontal direction compared to that during vertical jumps [17]. In sprint running, peak hip extension torque [18,19] and power [18] increases as running speed increases, whereas hip negative power is not observed. The purpose of sprint running is to achieve a high running speed (horizontal velocity). Therefore, the hip extensors are also important for maximizing horizontal speed (jump distance) in the double-leg rebound jump.

For the knee joint, unlike the hip joint, negative power increased during the 75% HDJ and longer HDJs, whereas positive power decreased (Table 2). In addition, as the jump distance increased, knee joint flexion torque was observed earlier (at approximately 80% of the take-off phase) and extension torque was smaller during knee extension (between approximately 40–80% of the take-off phase). Thus, the knee extensor primary function may have changed from a mechanical energy generator to an absorber during the 75% HDJ and longer jumps. During horizontal jump with double-leg take-off with counter movement, the lower knee joint kinetics control the jumping direction [15,16]. Additionally, in sprint running, knee extension is not necessary to achieve a high sprinting velocity, as faster sprinters show less knee extension during the support phase [20], and the knee joint may largely be a facilitator for the transfer of power, rather than the generator power, from the hip through the ankle during the support phase [28]. Moreover, in sprint running, knee extension velocity during the support phase becomes more stable as horizontal velocity increases [29]. Similar results were reported during bounding exercise with increasing jump step (increasing the horizontal velocity), which consists of a series of seven forward alternating single-leg jumps to cover the longest distance possible [14]. These results were reported for various horizontal movements. They are consistent with my observations on knee joint biomechanics and indicate that the knee joint controls the jumping direction in a double-leg rebound jump at higher horizontal velocity.

At the ankle joint, although negative power only varied with increasing jump distance, the magnitude of effect was low, and other variables did not change (Table 2). This is contrary to findings on other horizontal movements reported previously. In running at speeds of up to ~7 m/s and bounding, increasing horizontal velocity (jump distance) correlated with ankle joint kinetics [14,19]. This indicates that the ankle joint is important for achieving a faster horizontal speed. The VDJ is a repeated rebound-type jump (i.e., jumping as high as possible to minimize ground contact time) in the vertical direction with a double-leg take-off from a standing position. The ankle joint kinetic parameters (joint torque and power) during the VDJ are the highest among various types of jumps, including rebound-type drop jumps from a height of 0.3–0.5 m [10], counter movement jump types [9], and single-leg rebound jumps in the vertical direction [30]. Due to these characteristics of ankle joint kinetics during the VDJ, the ankle joint is important during the double-leg rebound jump in both the vertical and horizontal direction, which is contrary to that seen during other horizontal movements.

To the best of my knowledge, this is the first study investigating the effect of jump direction and jump distance on take-off legs joint kinetics during double-leg rebound jumps. The HDJ is often used for plyometric training (e.g., a drop jump in the horizontal direction, and the hurdle jump) as part of direction-specific training regimens. However, it remains unclear if direction-specific kinetics exist in the double-leg rebound jump. Based on my results, in the horizontal rebound jump using over 75% of the 100% HDJ distance, the hip and knee joint kinetic patterns are similar to those of other horizontal locomotor exercises, such as running and bounding [14,18,19,20,28,29]). Therefore, horizontal rebound jumps using jump distances of over 75% HDJ (approximately 1.66 ± 0.13 m; Table 1) would be more suitable for improving the hip and ankle joint power under similar kinetic patterns as horizontal locomotor exercises. This information would help elucidate direction-specific kinetics and facilitate efforts to effectively tailor plyometric exercises for various sports and the training needs of athletes involved in different sports.

There are certain limitations that must be acknowledged when interpreting the results of the present study. Firstly, because jump conditions affect the kinematics and kinetics (e.g., the height dropped from in the drop jump [6] and the hurdle height in the hurdle jump [31]), the results of this study cannot be extrapolated to other types of horizontal plyometric exercises with double-leg take-off. Secondly, results of the present study are limited in that there was a relatively small number of athletic participants; further, all participants were men. Therefore, my results may not be generalizable to women or non-athletic populations. Finally, my results are based on a cross-sectional study. A longitudinal study is warranted to elucidate the training effect of VDJ and HDJ.

## 5. Conclusions

Differences in joint kinetics between HDJ and VDJ were observed in the hip and knee joint kinetics and were associated with jump direction. Differences became more pronounced as jump distance increased.

## Figures and Tables

**Figure 1 sports-07-00183-f001:**
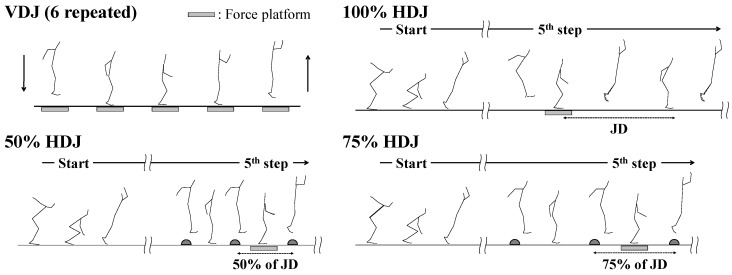
Experimental tasks in this study. VDJ, vertical double-leg rebound jump; 50% HDJ, horizontal double-leg rebound jump for 50% of the jump distance performed in 100% HDJ; 75% HDJ, horizontal double-leg rebound jump for 75% of jump distance performed in 100% HDJ; 100% HDJ, horizontal double-leg rebound jump; JD; jump distance in 100% HDJ.

**Figure 2 sports-07-00183-f002:**
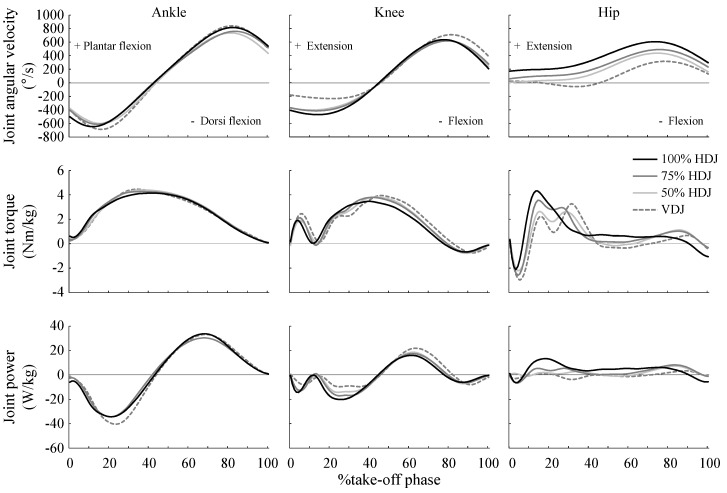
Average patterns of joint angular velocity, joint torque, and joint power for the ankle, knee, and hip joint. %take-off phase, normalized time series data to take-off phase time (0–100%); VDJ, vertical double-leg rebound jump; 50% HDJ, horizontal double-leg rebound jump for 50% of jump distance performed in 100% HDJ; 75% HDJ, horizontal double-leg rebound jump for 75% of jump distance performed in 100% HDJ; 100% HDJ, horizontal double-leg rebound jump.

**Table 1 sports-07-00183-t001:** Comparison of performance variables by jump distance (mean ± SD).

Performance Variables	VDJ	50% HDJ	75% HDJ	100% HDJ	Differences	Effect Size Partial η^2^
Jump distance (m)	–	1.11 ± 0.09	1.66 ± 0.13	2.21 ± 0.18	–	–
Contact time (s)	0.147 ± 0.019	0.148 ± 0.009	0.148 ± 0.014	0.147 ± 0.016	n.s.	0.008
Vertical velocity (m/s)	3.15 ± 0.20	3.06 ± 0.19	3.02 ± 0.21	2.70 ± 0.21	VDJ,50%,75% > 100%	0.682
Horizontal velocity (m/s)	−0.03 ± 0.18	1.56 ± 0.23	2.18 ± 0.30	3.11 ± 0.30	VDJ < 50% < 75% < 100%	0.990

VDJ, vertical double-leg rebound jump; 50% HDJ, horizontal double-leg rebound jump for 50% of jump distance performed in 100% HDJ; 75% HDJ, horizontal double-leg rebound jump for 75% of jump distance performed in 100% HDJ; 100% HDJ, horizontal double-leg rebound jump; α < 0.05.

**Table 2 sports-07-00183-t002:** Comparison of peak joint kinetics by jump distance (mean ± SD).

Joint Kinetics	VDJ	50% HDJ	75% HDJ	100% HDJ	Differences	Effect Size Partial η^2^
**Hip**						
Ecc torque (N·m/kg)	3.91 ± 1.88	3.67 ± 1.43	4.81 ± 2.05	5.91 ± 2.62	n.s.	0.356
Con torque (N·m/kg)	1.38 ± 1.09	1.49 ± 0.55	1.26 ± 0.49	1.10 ± 0.57	n.s.	0.093
Negative power (W/kg)	−4.40 ± 4.25	−0.83 ± 2.10	−0.33 ± 0.83	0 ± 0	VDJ > 50%,75%,100%	0.422
Positive power (W/kg)	4.19 ± 2.73	9.37 ± 2.89	11.15 ± 3.91	18.51 ± 9.83	VDJ < 50%,75% < 100%	0.588
**Knee**						
Ecc torque (N·m/kg)	4.23 ± 1.31	3.91 ± 1.14	4.20 ± 1.21	3.76 ± 1.11	n.s.	0.016
Con torque (N·m/kg)	4.00 ± 1.27	3.90 ± 1.01	3.84 ± 0.96	3.53 ± 1.19	n.s.	0.126
Negative power (W/kg)	−14.48 ± 7.67	−18.98 ± 7.13	−21.57 ± 8.54	−23.34 ± 12.13	VDJ < 75%,100%	0.307
Positive power (W/kg)	23.18 ± 9.01	18.83 ± 5.49	18.10 ± 5.77	16.27 ± 6.22	VDJ > 75%,100%	0.311
**Ankle**						
Ecc torque (N·m/kg)	4.57 ± 1.32	4.47 ± 0.92	4.39 ± 1.08	4.17 ± 1.28	n.s.	0.048
Con torque (N·m/kg)	4.38 ± 1.22	4.39 ± 0.88	4.36 ± 0.95	4.18 ± 1.08	n.s.	0.019
Negative power (W/kg)	−41.06 ± 12.66	−35.93 ± 10.92	−35.04 ± 11.17	−37.12 ± 20.11	VDJ > 50%,75%	0.088
Positive power (W/kg)	34.21 ± 12.20	32.23 ± 9.11	32.68 ± 9.34	34.29 ± 12.50	n.s.	0.022

VDJ, vertical double-leg rebound jump; 50% HDJ, horizontal double-leg rebound jump for 50% of jump distance performed in 100% HDJ; 75% HDJ, horizontal double-leg rebound jump for 75% of jump distance performed in 100% HDJ; 100% HDJ, horizontal double-leg rebound jump; Ecc, eccentric phase; Con, concentric phase; α < 0.05.

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
