# Peer review of "Effect of Jump Direction on Joint Kinetics of Take-Off Legs in Double-Leg Rebound Jumps"

_sports, 2019, doi:10.3390/sports7080183_

Round 1
Reviewer 1 Report
Thank you for this interesting study and the well-written manuscript.
Before I can recommend publication, please carefully consider the following issues:
Major issues:
lines: 57-63: Please also consider and discuss the role of the hamstring muscles commonly considered “knee flexors” as dominant muscles for horizontal speed generation in sprinting (Lombard’s paradox: The hamstring knee „flexors“ actually extend the knee for knee flexion angles lower than roughly 30°.
line 85: 30–60 min is a rather long period for warm-up, and a very broad range. After 60 min, I expect (probably uncontrolled) effects of fatigue, especially for sprinters. Why was the warm-up not standardized in a better fashion? How can the author nonetheless guarantee the comparability/homogeneity of obtained experimental data?
line 110: Ground reaction forces in 3D or 1D? What is meant with “one-force” platform? 3D information is required for a valid assessment of vertical and horizontal kinetics. Please give more precise information and critically discuss if only 1D data was used.
line 145: eta^2: Was an ANOVA part of the statistical analysis? If so, which I strongly advise, please provide details on that as well. t-tests on their own are not sufficient.
lines 200–222: Also discuss role of hamstring muscles in this context (see above).
Minor points:
line 14. hip negative power decreased. At what moment in time? At landing?
lines 14-15 : Please provide the units W/kg for all numbers, especially the first, i.e. “for -4.40+-4.25 W/kg” etc.
line 11: recommend renaming "HDJ" to "100% HDJ" for terminological consistency and better understanding throughout the manuscript
lines 14-19: The sentences are written in a too cumbersome manner to easily understand them. I advise to rephrase for better understanding according to the scheme “hip negative power decreased from 100 % via 50 % to 25 % HDJ ()” etc.
lines 35 and 42: It is unclear to the reader if the author addresses double-leg jumps (which then is take-off leg?) or single leg jumps here. Please clarify.
line 46: “during take-off leg”: Bad English; better: “in take-off leg”
line 52-53: As before, what is take-off leg for double-leg jumps? I assume both are meant, so please use plural: “take-off legs”
line 77: (Figure 1 - nicely done!): Please rename HDJ to 100% HDJ for better terminological consistency (see above).
line 112: Please give details on the humanoid model: Standard model or specific (custom-built) software solution?
line 126 “based on 3% of the body mass” unclear statement. Is the threshold of ground reaction force level meant? Please give a more precise wording.
line 131: Were the participants all Japanese? If so, please give that important information in the methods section above.
line 152: Unit for reactive strength index (m/s as stated in the Methods section!?) is missing.
lines 154–167: Please give p values for your comparisons. Naturally, p<0.05 is understood with your comment in the Methods sections, but explicit values would be beneficial for the reader.
line 252: Rcareplace “there were” by “there was”
line 255: Insert “A”: “A longitudinal study…”
Author Response
RESPONSE TO REVIEWER 1:
We wish to express our strong appreciation to the Reviewer for his or her insightful comments on our paper. We feel the comments have helped us significantly improve the paper.We wish to express our strong appreciation to the Reviewer for his or her insightful comments on our paper. We feel the comments have helped us significantly improve the paper.We wish to express our strong appreciation to the Reviewer for his or her insightful comments on our paper. We feel the comments have helped us significantly improve the paper.We wish to express our appreciation to the Reviewer for his or her insightful comments, which have helped us to significantly improve the paper.
Comments 1:
Major issues:
lines: 57-63: Please also consider and discuss the role of the hamstring muscles commonly considered “knee flexors” as dominant muscles for horizontal speed generation in sprinting (Lombard’s paradox: The hamstring knee „flexors“ actually extend the knee for knee flexion angles lower than roughly 30°.
Response: I thank the reviewer for this comment. This study could not investigate and discuss the Lombard’s paradox because EMG data was not recorded. Therefore, I was unable investigate the role of the hamstring muscles in this study, which used only joint kinetics. Moreover, the Lombard’s paradox is observed while rising to cycle or while standing from a sitting or squatting position. However, there is no evidence regarding Lombard’s paradox in the jump movement. Therefore, I was unable to consider and discuss the Lombard’s paradox during this revision. I would be grateful if you could explain why and how the Lombard’s paradox is applicable in this study. I apologize for bothering you.
Comments 2:
line 85: 30–60 min is a rather long period for warm-up, and a very broad range. After 60 min, I expect (probably uncontrolled) effects of fatigue, especially for sprinters. Why was the warm-up not standardized in a better fashion? How can the author nonetheless guarantee the comparability/homogeneity of obtained experimental data?
Response: In this study, the warm-up session was not standardized because the most suitable warm-up (time and content) varies according to the individual. Moreover, the participants in this study were athletes, and their warm-up in the T&F meeting is customized and hence could not be standardized. Therefore, the study included this warm-up style. I have revised this in the manuscript (Lines 84-87; “Track Changes” Lines 90-93).
Comments 3:
line 110: Ground reaction forces in 3D or 1D? What is meant with “one-force” platform? 3D information is required for a valid assessment of vertical and horizontal kinetics. Please give more precise information and critically discuss if only 1D data was used.
Response: All ground reaction forces data are presented as 3D in my study. “One-force platform” implies that the number of force platforms is 1. I have corrected this in the revised manuscript (Lines 109-112; “Track Changes” Lines 116-119).
Comments 4:
line 145: eta^2: Was an ANOVA part of the statistical analysis? If so, which I strongly advise, please provide details on that as well. t-tests on their own are not sufficient.
Response : I thank the reviewer for this comment. In accordance with the reviewer's comment, I have added the results of ANOVA in the revised manuscript (Lines 155-173; “Track Changes” Lines 163-181).
Comments 5:
lines 200–222: Also discuss role of hamstring muscles in this context (see above).
Response: This study could not investigate and discuss the Lombard’s paradox because EMG data was not recorded. Therefore, I was unable investigate the role of the hamstring muscles in this study, which used only joint kinetics. Moreover, the Lombard’s paradox is observed while rising to cycle or while standing from a sitting or squatting position. However, there is no evidence regarding Lombard’s paradox in the jump movement. Therefore, I was unable to consider and discuss the Lombard’s paradox during this revision. I would be grateful if you could explain why and how the Lombard’s paradox is applicable in this study. I apologize for bothering you.
Comments 6:
Minor points:
line 14. hip negative power decreased. At what moment in time? At landing?
Response: The hip negative power decreased during the eccentric phase (0-10% of %take-off phase). Accordingly, I have added this in the revised manuscript (Lines 13-14; “Track Changes” Line 15).
Comments 7:
lines 14-15 : Please provide the units W/kg for all numbers, especially the first, i.e. “for -4.40+-4.25 W/kg” etc.
Response: In accordance with the reviewer's comment, I have revised the manuscript based on words limit (200 words) (Lines 14-19; “Track Changes” Lines 15-21).
Comments 8:
line 11: recommend renaming "HDJ" to "100% HDJ" for terminological consistency and better understanding throughout the manuscript
Response: In accordance with the reviewer's comment, I have revised this throughout this manuscript.
Comments 9:
lines 14-19: The sentences are written in a too cumbersome manner to easily understand them. I advise to rephrase for better understanding according to the scheme “hip negative power decreased from 100 % via 50 % to 25 % HDJ ()”
etc.
Response: In accordance with the reviewer's comment, I have revised this (Lines 13-19; “Track Changes” Lines 15-21).
Comments 10:
lines 35 and 42: It is unclear to the reader if the author addresses double-leg jumps (which then is take-off leg?) or single leg jumps here. Please clarify.
Response: In accordance with the reviewer's comment, I have corrected this in the manuscript.
Comments 11:
line 46: “during take-off leg”: Bad English; better: “in take-off leg”
Response: In accordance with the reviewer's comment, I have corrected this in the manuscript (Line 46; “Track Changes” Line 49).
Comments 12:
line 52-53: As before, what is take-off leg for double-leg jumps? I assume both are meant, so please use plural: “take-off legs”
Response: In accordance with the reviewer's comment, I have corrected this in the manuscript.
Comments 13:
line 77: (Figure 1 - nicely done!): Please rename HDJ to 100% HDJ for better terminological consistency (see above).
Response: I thank the reviewer for this encouraging comment. In accordance with the reviewer's comment, I have corrected this throughout this manuscript.
Comments 14:
line 112: Please give details on the humanoid model: Standard model or specific (custom-built) software solution?
Response: This is the standard model of Ae [22]. Therefore, I have added this information in the manuscript (Line 113; “Track Changes” Line 120).
Comments 15:
line 126 “based on 3% of the body mass” unclear statement. Is the threshold of ground reaction force level meant? Please give a more precise wording.
Response: In accordance with the reviewer's comment, I have revised this as “the threshold of the ground reaction force level was based on 3% of the body mass” (Line 128; “Track Changes” Line 135).
Comments 16:
line 131: Were the participants all Japanese? If so, please give that important information in the methods section above.
Response: In accordance with the reviewer's comment, I have revised the sentence as “Twelve Japanese male track and field sprinters and jumpers (age, 22.0±2.2 years; height, 1.75±0.61 m; mass, 65.80±4.02 kg) participated in this study.” (Line 66; “Track Changes” Line 70).
Comments 17:
line 152: Unit for reactive strength index (m/s as stated in the Methods section!?) is missing.
Response: In accordance with the reviewer's comment, I have added the unit (Line 159; “Track Changes” Line 167).
Comments 18:
lines 154–167: Please give p values for your comparisons. Naturally, p<0.05 is understood with your comment in the Methods sections, but explicit values would be beneficial for the reader.
Response: In accordance with the reviewer's comment, I have added p values in the Appendix (Table A1) because of much information about p values in the revised manuscript.
Comments 19:
line 252: Rcareplace “there were” by “there was”
Response: In accordance with the reviewer's comment, I have revised this (Line 268; “Track Changes” Line 277).
Comments 20:
line 255: Insert “A”: “A longitudinal study…”
Response: In accordance with the reviewer's comment, I have revised this (Line 271; “Track Changes” Line 280).
Thank you again for your comments on my manuscript, which have helped improved it.
Reviewer 2 Report
Effect of jump direction on take-off leg joint kinetics in double-leg rebound jumps
Manuscript # - Sports-543179
General Comments:
The paper is well written and addresses kinetic differences between two types of rebound jumps. The ethics board approval and informed consent process has been described in the manuscript. The introduction, methods, results and discussion are well addressed. Statistical analyses is okay and suffices the current data set. The only big addition that I would suggest is for the data analysis, as described below.
Other minor comments, questions and edits are listed under specific comments.
The procedures are explained well and all data analysis was explained in periodically. However, for the data analysis section, I highly suggest that the author adds the equations for calculations for the inverse dynamics approach, in addition to the description already provided. This will benefit the readers and other biomechanists, kinesiologists and exercise scientists to recreate the study and its analysis.
The statement about Professor Koji Zushi is a very nice gesture. Thank you.
Please add the data analysis in equation form in addition to the description in the data analysis.
Specific Comments:
Line 10-11: Include the other jumps here in this sentence.
Please provide rationale for using 50% and 75% jump heights and distances. Importance of using these percentage jumps in training and how it can impact competition athletic performance.
Were there familiarization trials performed, even though all athletes had 7 years of prior practice as stated in the manuscript?
Provide rationale for using only 12 participants.
Provide rationale for using only male population.
Author Response
RESPONSE TO REVIEWER 2:
We wish to express our strong appreciation to the Reviewer for his or her insightful comments on our paper. We feel the comments have helped us significantly improve the paper.We wish to express our strong appreciation to the Reviewer for his or her insightful comments on our paper. We feel the comments have helped us significantly improve the paper.We wish to express our strong appreciation to the Reviewer for his or her insightful comments on our paper. We feel the comments have helped us significantly improve the paper.We wish to express our appreciation to the Reviewer for his or her insightful comments, which have helped us to significantly improve the paper.
Comments 1:
General Comments:
The procedures are explained well and all data analysis was explained in periodically. However, for the data analysis section, I highly suggest that the author adds the equations for calculations for the inverse dynamics approach, in addition to the description already provided. This will benefit the readers and other biomechanists, kinesiologists and exercise scientists to recreate the study and its analysis.
Response: I thank the reviewer for this comment. In accordance with the reviewer's comment, I have added this in Appendix (Figure A1) (Line 282; “Track Changes” Line 291).
Comments 2:
Specific Comments:
Line 10-11: Include the other jumps here in this sentence.
Response: I thank the reviewer for this comment. Because the jumps were not defined previously, the definitions are not included in sentence and have been mentioned in the subsequent sentence.
Comments 3:
Please provide rationale for using 50% and 75% jump heights and distances. Importance of using these percentage jumps in training and how it can impact competition athletic performance.
Response: The results of this study can make athletes select the training intensity (jump distance) based on the individual jump performance. This is the importance of using percentage (50% and 75% HDJ) jumps in plyometric training.
Comments 4:
Were there familiarization trials performed, even though all athletes had 7 years of prior practice as stated in the manuscript?
Response: Participants were familiar with all experimental trials, because they had performed regular sprint and jump training, including experimental tasks, 5 to 6 days per week for more than 7 years. Therefore, I have added this in the revised manuscript (Line 71-72; “Track Changes” Lines 75-76).
Comments 5:
Provide rationale for using only 12 participants.
Response: As reviewers mentioned, this is a limitation of this study. I could not recruit more participants because other athletes were injured in our team. Therefore, this has reported as the limitation in the revised manuscript (Line 268-270; “Track Changes” Line 277-279).
The Shapiro-Wilk test was used to test for normal distribution of the reactive strength index in the VDJ and 100% HDJ. As the results, these values had normal distribution with the Shapiro-Wilk test (VDJ, p = 0.323; 100% HDJ, p = 0.760). I have added this to provide the rationale (Line 152-153, 157-158; “Track Changes” Lines 161-162, 165-166).
Comments 6:
Provide rationale for using only male population.
Response: As mentioned by the reviewer, this is the limitation of this study. I could not recruit women athletes. Therefore, the present results may not be generalizable to women. This has been mentioned in the limitation section (Lines 168-270; “Track Changes” Lines 277-279).
Thank you again for your comments on my manuscript, which have helped improved it.